# Preparation and Properties of Bulk and Porous Ti-Ta-Ag Biomedical Alloys

**DOI:** 10.3390/ma15124332

**Published:** 2022-06-18

**Authors:** Grzegorz Adamek, Mikolaj Kozlowski, Adam Junka, Piotr Siwak, Jaroslaw Jakubowicz

**Affiliations:** 1Institute of Materials Science and Engineering, Poznan University of Technology, Jana Pawla II 24, 61-138 Poznan, Poland; mikolaj.j.kozlowski@doctorate.put.poznan.pl (M.K.); jaroslaw.jakubowicz@put.poznan.pl (J.J.); 2Department of Pharmaceutical Microbiology and Parasitology, Wroclaw Medical University, Borowska 211, 50-534 Wroclaw, Poland; adam.junka@umed.wroc.pl; 3Institute of Mechanical Technology, Poznan University of Technology, ul. Piotrowo 3, 60-965 Poznan, Poland; piotr.siwak@put.poznan.pl

**Keywords:** titanium alloys, mechanical alloying, biomaterials, hot pressing, space holder

## Abstract

The paper presents the results of the preparation of bulk and porous Ti-Ta-Ag alloys. The first step of this study was the preparation of the powder alloys using mechanical alloying (MA). The second was hot-pressing consolidation and sintering with a space holder, which resulted in high-density and high-porosity (approximately 70%) samples, respectively. Porosity, morphology, mechanical properties, biocompatibility, and antibacterial behavior were investigated and related to the preparation procedures. The authors found that Ta and Ag heavily influence the microstructure and determine other biomaterial-related properties. These new materials showed positive behavior in the MTT assay, and antibacterial properties. Such materials could find applications in the production of hard tissue implants.

## 1. Introduction

Titanium alloys are most widely used in hard tissue implant applications owning to their high corrosion resistance, self-passivation, and great mechanical strength-to-density ratio compared to other metallic alloys. Moreover, in a group of materials used for permanent implants, they display the lowest Young’s modulus but still retain a high elastic modulus (65–110 GPa) in comparison with bone (10–30 GPa) [1,2,3,4], which can result in bone resorption, potentially leading to implant loosening or bone fracture in the worst scenario. There are two ways to reduce the elastic moduli. The first is the appropriate combination of chemical composition and the second and more effective is the preparation of a special form of a high porosity material referred to as metallic foams. Moreover, such porosity could act like a scaffold for the tissue and enhances body fluid transportation, resulting in great tissue–implant interconnection [5,6,7,8,9,10]. Newly designed alloys must contain additives that are also biocompatible and provide improvement in mechanical, chemical, and biological properties. Such elements are Zr, Nb, and Ta [11,12,13,14,15,16,17,18,19,20,21,22,23,24,25,26,27,28]. They belong to a non-toxic and non-allergic group of elements [29,30]. They should stabilize the beta titanium structure showing better Young’s moduli. Tantalum is considered to be one of the strongest beta stabilizers [29,30]. With the increasing concentration of Ta, the elastic moduli decreases while the strength increases [18,22,27,31] and the corrosion resistance improves compared to pure titanium, which is a result of higher stability of Ta_2_O_5_ over TiO_2_ [32,33]. The antibacterial behavior of biomaterials is also important and has been constantly improved. Silver is an element well-known for its antibacterial properties. The addition of a few (wt%) Ag into the alloy helps to keep the area of implantation free of bacteria and prevents biofilm formation [34,35,36]. That allows for the lowering of the total cost of treatment and mortality caused by infection and improves the patient’s comfort. Ti-Ta-Ag ternary alloys appear to be attractive from a biomaterials point of view; however, the information about them in the literature is insufficient. An example of such a study was shown by Zhang et al. They investigated Ti-25Ta-xAg (x = 1.5, 3 and 4.5) alloys prepared by a spark plasma sintering technique. The microstructure of the alloys was composed of Ti-a, Ti-a”, Ti-b, and Ag precipitates. The authors reported that Ag can significantly enhance the mechanical properties [36].

Titanium-based highly porous materials are difficult to process through liquid metallurgy due to Ti’s chemical reactivity and a relatively high melting point (1668 °C [37]. The common technique of preparation of metallic foams is sintering with a space holder material. The space holder particles are mixed with the metallic powder, then compacted and eventually removed during or before sintering. There are a number of different space holders investigated by many researchers, i.e., sucrose, sodium chloride, sodium fluoride, ammonium hydrogen carbonate, titanium hydride, magnesium, tapioca, and carbamide [38,39,40,41,42,43,44]. The advantages of using space holders are that the pore size, level of porosity, and pore morphology can be designed by varying the type, amount, and quality of the space holder. The carbamide has great potential in the preparation of titanium-based foams [45,46,47,48,49].

The main goal of this study is to investigate the effect of different contents of (30, 40 wt%)Ta with small amounts (3, 5 wt%) of Ag in titanium ternary alloys, as well as the preparation parameters (mechanical alloying time, sintering time, and temperature) on the microstructure, mechanical properties, corrosion resistance, and biological behavior. The authors investigated both bulk and porous samples prepared from mechanically alloyed powders. The authors believe that the study will expand the knowledge of Ti-Ta-Ag ternary alloys.

## 2. Materials and Methods

### 2.1. Materials Preparation

The ternary Ti-Ta-Ag alloys were prepared by mechanical alloying (MA) under argon atmosphere using a SPEX 8000 Mixer Mill. Based on our previous initial studies and literature, we decided to investigate the following chemical compositions: Ti-xTa-yAg, where x = 30, 40 wt% and y = 3, 5 wt%. Powders for MA were: Ti (purity 99.5%), Ta (99.98%) from AlfaAesar, and Ag (99.8%) from Aldrich, all 325 mesh. The weight of the starting powder was measured by precision balance (0.001 g repeatability; Radwag). The ball (steel) to powder weight ratio was 10:1. After a certain time (every 2 or 4 h), the milling process was paused to investigate the structural and powder morphology changes, which were performed by X-ray diffraction (XRD) and scanning electron microscopy (SEM). The milling time was designed to achieve a fine Ti-β structure, and it was 24 and 14 h, depending on the chemical composition of the alloys. Subsequently, the as-prepared powder alloys were consolidated by hot pressing combined with high-frequency induction heating (Elbit FI-W). The parameters of hot pressing were designed to achieve the highest possible density level and were as follows:Sintering temperature: 850 °C,Applied pressure: 50 MPa,Die material: graphite,Vacuum condition: <50 Pa,Heating rate: 85 °C/s,Hold time 10 s,Cooling: together with the furnace.

The compacts were 8 mm in diameter and approximately 4 mm in height. The samples for all tests were mechanically ground on the SiC papers and polished in a diamond suspension. To produce foam, the mechanically alloyed powders were dry-mixed with a space holder agent. In this work, the authors chose urea (ACS reagent from Sigma-Aldrich, Saint Louis, MO, USA) in the amount of 70 vol.%. After uniform mixing, the compacted parameters were as follows:Cold compaction: uniaxial pressing (8 mm diameter cylindrical steel die),Applied pressure: 1000 MPa,Time: 10 s.

The sintering was carried out in a vacuum tube furnace (Nabertherm, Germany) in two steps:


Preheating: (first step-carbamide removal) temp.: 300 °C,Hold time 10.8 ks,Sintering (straight after preheating) temp.: 1300 °C,Hold time 3.6 ks,Cooling: together with the furnace.


### 2.2. Structural Analysis

The structure and phase analysis were investigated using Empyrean XRD (Panalytical, Malvern, UK) with CuKα radiation (λ = 1.5418 Å) lamp equipped with a 0.04 rad soller slits, a 10 mm incident beam mask, a 1° anti-scatter slit, and 1/2° divergence slit in the incident beam. For the diffracted beam path, a 0.04 rad soller slit, a nickel beta-filter, and a 5.5 mm anti-scatter slit were used. Measurements were carried out at the following parameters: voltage 45 kV, anode current 40 mA, scanning range 30–80°, time per step 59.69 s/step, and step size 0.0334°. Crystallographic data and phase composition were calculated using Rietveld refinement by HighScore Plus software (Malvern Panalytical). The peaks profile analysis was carried out using a pseudo-Voigt profile function. For crystallite size and microstrain calculations, Scherrer’s equation was used. The phase composition for hot-pressed samples was evaluated by a standardless X-Ray diffraction method. The simulations were performed with selected models for Ti(α)–(ref. code 04-008-4973; space group-P63/mmc), Ti(β)–(ref. code 04-003-7297; space group-Im-3m), Ta–(ref. code 04-004-3017; space group Im-3m), and Ag–(ref. code 01-071-4613; space group-Fm-3m).

### 2.3. Microstructural and Morphological Analysis

The microstructure and morphology were characterized by Mira 3 FEG SEM (Tescan, Brno, Czech Republic) equipped with in-beam secondary electron (SE), backscattered electrons (BSE), and an energy-dispersive X-ray spectroscopy (EDS) UltimMax 65 (Oxford Instruments, Abingdon, UK) detector. The porosity level was examined by an GX51 (Olympus, Tokyo, Japan) optical microscope on polished samples, and the micrographs were analyzed using Stream Start and GIMP software through color histogram analysis. The grain size was measured by the Snyder–Graff method.

### 2.4. Corrosion Resistance Analysis

The corrosion tests of the materials were performed in the Ringer’s solution (BTL Zakład Enzymow i Peptonow, Lodz, Poland) using Solartron 1285 (Solartron Analytical, Farnborough, UK) potentiostat working in the potentiodynamic mode with a scan rate of 0.5 mV/s, in the established potential range of −0.7 V to +2.5 V vs. open circuit potential (OCP) and EK&G corrosion cell. The counter and reference electrodes were made from graphite and Pt, respectively. Before starting the potentiodynamic tests, the OCP was stabilized. The tests were performed at a temperature of 37 °C.

### 2.5. Wetting and Surfaces Free Energy Analysis

The wetting and surface free energy measurements were carried out on the Drop Shape Analyzer DSA25 (KRÜSS-Scientific, Hamburg, Germany). A distilled water and glycerol drop of 2.0 µL in volume was used for the analysis. The surface free energy was determined by measuring the contact angle (CA) according to the Owens, Wendt, Rabel, and Kaelble (OWRK) model by ADVANCE software. Other parameters were as follows: dosing speed 0.2 mL/min, measuring time 5 s, probing frequency for multiple measurements 10 fps, Young’s Laplace CA fitting method, and ambient temperature 22.0 °C.

### 2.6. Nanoindentation and Mechanical Tests

The mechanical measurements were made with a Picodentor HM500 (Helmut Fisher) nanoindenter. According to the ISO 14577-1 [50] standard, the following parameters were measured: the HM-Martens Hardness, the HV-Vickers Hardness, and the EIT-indentation modulus. The indentations were made at a force of 300 mN for 20 s. The mechanical properties of the porous counterparts were measured by a compression test. A 4483 Instron mechanical testing machine with a strain rate of 0.001 was used to measure the compressive strength and the elastic moduli.

### 2.7. MTT Assay

Normal human osteoblast (CC-2538) (NHost) and human periodontal ligament fibroblasts (CC-7049) (HPLF) (Lonza Group Ltd., Basel, Switzerland) were used for the in vitro cytocompatibility tests, obtained under static conditions. The cells were cultured at a concentration of 5000 cells/well in 1 mL of culture medium on each sample at 37 °C in a 5% CO_2_ incubator for 24, 48, 72, and 96 h. The proliferation of the cells in the conditioned mediums was expressed as a percentage of the value of relative viability of the cells (RVC) of the reference medium. The reference medium was prepared using pure bulk microcrystalline cp-titanium samples, and in the graph it is represented by the value of 100%. All samples were sterilized by autoclaving at 120 °C for 15 min and separately located in 24-well microplates. The statistical significance was analyzed using the Kruskal–Wallis OneWay Analysis of the Variance on Ranks with the multiple repetition option SigmaStat 3.5 (Systat Software Inc., Erkrath, Germany) with U-Mann Whitney test. The significance level was *p*-value < 0.05.

### 2.8. Antibacterial Properties

The antibacterial properties were examined with the following collection of strains (from ATCC): *S. aureus* 6538; *P. aeruginosa* 15442; and *C. albicans* 10231. During biofilm formation experiments, all strains were cultured on trypticase soy broth (TSB) liquid media (Biocorp, Warsaw, Poland), and appropriate agar (Columbia, Sabouraud, McConkey, Biocorp, Poland) media were used during the preservation of the strains. The alloys and cp-Ti in the form of discs were the study and the control group, respectively. The strains cultured on appropriate agar plates (*S. aureus*, Columbia plate; *C. albicans*, Sabourad plate; and *P. aeruginosa*, McConkey plate) were transferred to a liquid TSB medium and incubated for 24 h at 37 °C under aerobic conditions. After incubation, the strains were diluted to the density of 1 McFarland (MF) (densitometer Densi-LA-meter II, Biosciences, San Francisco, CA, USA). The microbial dilutions were inoculated to wells of 24-well plates containing titanium alloy discs. Another 24 h/37 °C incubation was performed. Next, the discs were rinsed using sterile saline to remove the unbound cells and to leave the biofilm on the disc surface only. Afterwards, the discs with the biofilm were transferred to a fresh, sterile TSB medium containing 1% 2,3,5-triphenyl-2H-tetrazolium chloride (TTC. Sigma-Aldrich, Saint Louis, MO, USA) and left for 4 h. TTC is a colorless compound that changes into red formazane in the presence of living, metabolically active microbial cells. After incubation, the discs were transferred to an ethanol-acetic acid mixture 95:5 (*vol*/*vol*) and shaken vigorously on the plate-shaker (Schuttiken, Germany) to release dye from inside of the cells. Subsequently, the medium suspended in formazane was collected and its concentration was measured using a spectrometer (Thermo Scientific Multiscan GO, Waltham, MA, USA) at a wavelength of 480 nm. Statistical analysis was performed with the SigmaStat package, Version 2.0 (SPSS, Chicago, IL, USA). Quantitative data from the experimental results were analyzed using an unpaired *t*-test with Welch’s correction, with the significance level established at *p* < 0.05.

## 3. Results and Discussion

In this paper, the Ti-Ta-Ag alloys were investigated. Mechanical alloying (MA) and powder metallurgy (PM) were used to develop the materials in the form of bulk and highly porous counterparts. The first step of the preparation of the alloys was MA, and the structural changes during the process were controlled by XRD. In Figure 1, the XRD data have been shown for all the prepared alloys. Based on the diffractograms, some trends can be described. First of all, for all the chemical compositions we can observe the phase transformation as a result of the alloying elements dissolving in the Ti structure. Ta is a strong β-Ti stabilizer, and the results confirm that alloys with a higher Ta content (40%) need a significantly shorter time of 14 h to achieve beta structure, which is 10 h and over 40% less than alloys with a 30% Ta content. In all the cases, the amount of Ag was sufficiently low to make the peaks disappear after 2 h of milling. The morphology of the powders (SEM-BSE images in Figure 1) was also investigated. We can see that the powders after 2 h of MA are just mixtures of elements. The BSE contrast that indicates the presence of heavy (bright) and light (dark) elements in certain areas is not visible for the final powder agglomerates, which confirms good distribution. For all the alloys, the particle size, measured with Tescan SEM software decreased during milling to reach approximately 60 μm for Ti30Ta3Ag and approximately 20 μm for the other alloys.

The crystallite size and strain after MA were calculated using the Williamson–Hall (W-H) formula. The results have been presented in Table 1. We can find that there is an increase in the strain value with the increasing Ag from 3 to 5. It shows that the Ag content could have an influence on the crystallographic parameters.

Figure 2 shows the microstructure of hot-pressed titanium alloys. On the BSE images, the two-phase microstructure is clearly visible. The microstructure is homogenous, consisting of fine regular grains. The bright areas represent Ta-rich Ti-β, while the dark regions represent the Ti-α microstructure. There is a significant difference in the grain size among the alloys. The average grain size was in the range 200–300 nm and 200–400 nm for Ti30Ta3Ag and Ti30Ta5Ag, respectively. For alloys with 40% Ta content, the same trend was observed. The average grain size for Ti40Ta3Ag and Ti40Ta5Ag was in the range 50–100 nm and 70–200 nm, respectively. The increasing silver content results in larger grains. However, the Ta content has an opposite influence. The alloys with 40% of Ta have, on average, four times smaller grains despite their shorter milling time and smaller crystal size after MA.

Using hot-pressing and optimized sintering parameters, there is a possibility of achieving high-density compacts. The porosity of hot-pressed alloys was as follows: 0.03, 0.03, 0.05, and 0.07% for Ti30Ta3Ag, Ti30Ta5Ag, Ti40Ta3Ag, and Ti40Ta5Ag, respectively, which is a very good result. The pores could negatively influence the mechanical parameters; hence, when developing bulk materials, it is important to reduce their size. It seems to be a trend that the porosity increases with the increasing content of the alloying elements; yet, it still remains on a low level.

Figure 3 shows the XRD spectra of the hot-pressed alloys. Every alloy exhibited a two-phase structure of Ti-β and Ti-α, which confirms the microscopic results. With the increasing Ta content, the more dominant peaks of Ti-β are visible, while the intensity of Ti-α peaks decreases. There is also a significant influence of Ag on the stabilization of the titanium structures. The crystallographic data have been calculated with Rietveld refinement using High-Score Plus software and complied in Table 1, along with the phase composition.

With the increasing Ta and Ag concentration, the volume of Ti-β increases. Moreover, in alloys with 5% silver, the extra peaks are observed on diffractograms, which corresponds to the content of Ag. The solubility of Ag in titanium is relatively low, and in the room temperature it reaches approximately 2–3%; thus, we could expect silver precipitates in the microstructure. They are visible on XRD; however, because of their low amount (1.6 and 2.3%) and size, we could not record them on the SEM micrographs. Zhang et al. has shown that for Ti-25Ta-(1.5, 3, 4.5)Ag alloys prepared by SPS, the Ag precipitates are located at the grain boundaries. They sintered the alloys for 5 min, which is 30 times longer than the samples in this work. Such time results in a relatively greater size of the Ag precipitates (approximately 2–3 μm) [8]. Additionally, in our study there was no defined influence of the investigated alloying elements concentration on the lattice parameters. The calculated lattice parameters of Ti-α and Ti-β are higher or close to the reference values (Ti-α: a = 2.951 Å, c = 4.684 Å; and Ti-β: a = 3.276 Å) [51,52].

The nanoindentation (load-displacement) curves and the collected results have been shown in Figure 4 and Table 2. The chemical composition and the microstructure heavily influence the mechanical properties. From the data, we can clearly see that the increasing amount of Ta increases the hardness. However, a higher concentration of Ag leads to its decrease. When it comes to the Young’s moduli, there is a significant decrease in their values with an increase in the amount of the alloying elements, which also corresponds to a higher beta phase concentration. However, the Young’s moduli are still quite high and correspond to microcrystalline Ti. Based on the literature, the Ti-Ta alloys should be characterized by much lower moduli. Zhou et al. show that the Young’s moduli for Ti-30Ta, Ti40Ta, and Ti-70Ta were 69, 83, and 67 GPa, respectively [22,23]. The reason for such mechanical properties obtained in this work could be very fine the microstructure, as well as the solid-solution strengthening. Another way to decrease the elastic moduli is to prepare porous materials, which is the next step of this study.

Another important property from the biomaterials point of view is the corrosion resistance investigated in the Ringer’s solution using the potentiodynamic method. The corrosion curves are shown in Figure 5 (parameters in Table 3). For all the tested materials the corrosion potential (E_corr_) was at approximately −0.32 to over −0.46 V. The highest value of E_corr_ was recorded for Ti30Ta5Ag. However, the values for all the alloys in general appear to exhibit a trend that the corrosion resistance increases with the increasing content of the alloying elements (both Ta and Ag). The impact of Ta is more strongly highlighted than that of Ag. Additionally, the alloys with a higher Ta concentration show a wider plateau range; therefore, we could expect a greater passivation tendency.

Wetting properties have a great impact on hard tissue growth due to their influence on biofilm formation. Wetting properties have been shown in Table 4, where some differences among the investigated alloys can be observed. It cannot be concluded that the chemical composition has a direct influence on the wetting properties. However, the contact angle, as well as the surface free energy, explicitly depend on the grain size. The contact angle increases and SFE decreases with the decreasing grain size. The highest SFE was observed for Ti30Ta3Ag (with the grain size in the range 200–400 nm) and the lowest for Ti40Ta3Ag and Ti40Ta5Ag (with the grain size in the range 50–100 and 70–200 nm, respectively). For all the samples, the contact angle is significantly below 90°; hence, all the samples exhibit moderately hydrophilic properties despite the fact that the investigated materials have a polished surface. The properties could be additionally improved by surface modification.

The preparation and characterization of the bulk materials was the first phase of the current project. The second phase was the fabrication of the porous counterparts from mechanically alloyed powders. In the current study, the authors decided to use urea (carbamide) as the space holder material. Figure 6 shows an example of the SEM micrographs of a Ti40Ta3Ag broken foam sample. In Figure 6a,b, we can observe a general view and higher magnification, respectively. The micrographs in Figure 6c,d present the pore wall morphology, which is also well developed. There are two kinds of pores visible at different magnification levels. In Figure 6a,b, we can see many relatively large (10 to over 200 μm), well interconnected, and irregularly shaped pores. The other pores are much smaller (approximately 0.25 to 1 μm) and are located in larger pore walls. They are interconnected to a lesser extent. In Figure 6d, we can see the grain morphology of the sintered foam. It consists of fine grains in the range of 200–300 nm. The total porosity is approximately 70%, which nicely corresponds with the designed value. The shrinkage level is very low. When compared with the studies of other authors, the results look very good [46,47,48,49]. Guibao et al. investigated titanium foams prepared with the urea spacer (powders prepared by mortar mixing for 30 min), and they achieved 56.23% porosity (with 70 vol% of urea) [49]. Tuncer et al. prepared their foams with 8–12% shrinkage, and they mixed the powders by ball milling for 30 min. [48]. Dry mixing in this study takes a few minutes. This, as well as the application of mechanically alloyed powders, could be the reason for lower shrinkage because the mass transportation in such fine powders during sintering going through the grain boundaries rather than the pore walls results in metallurgical bonding. The pore size distribution has been shown in Figure 7. The trend line confirms the existence of two types of pores, and the relation of the larger to the smaller is almost seven times in favor of the larger ones.

The mechanical properties of the prepared foams were measured by the compression test, and the results have been shown in Table 5. The compression strength of all the alloys is in the range 15 to over 17 MPa and the elastic moduli oscillates around 1 GPa. The results show the great potential influence of porosity on the reduction of the Young’s moduli and a negligible influence of the chemical composition on the mechanical properties in such highly porous materials. The moduli are close to that of the human cancellous bone (0.1 to 2 GPa), thus increasing the application potential of the discussed materials [10].

The cytotoxic activity was analyzed using the MTT assay. The obtained RVC results for normal human osteoblasts and human periodontal ligament fibroblasts have been shown in Figure 8a,b, respectively. The results revealed a significant effect of time on the cell viability. After 24 h, the RVC for all the tested materials was lower (oscillating in the range 57–60%) than that of the reference sample, which is a frequent situation caused by the longer adaptation time of the cells to the richer chemical composition of a new medium. The cytotoxicity decreased over time. Such results allow for us to conclude that all the tested alloys are non-toxic.

Moreover, there were no significant differences for each chemical composition. At 96 h, all the materials showed the highest viability. For the fibroblasts (HPLF), the alloys showed a similar RVC (approximately 1.2 times better). The results for the osteoblasts (NHost) showed the same trend; however, the RVC was more varied. The best results of 1.21 RVC were recorded for Ti30Ta3Ag and 1.12 RVC for Ti40Ta5Ag. In all the cases, the new materials showed better behavior compared to bulk cp-Ti.

The antibacterial behavior of biomaterials is an important factor currently investigated, and silver is the most important element in the design of such properties. In this work, the authors investigate alloys consisting of 3 and 5% of Ag. As it can be observed in Figure 9, the biofilm of all the tested microbial species formed less eagerly on the surface of Ag-doped samples compared to cp-Ti. In the case of the biofilm formed by *S. aureus*, *P. aeruginosa*, and *C. albicans*, these differences were statistically significant (*t*-test, *p* < 0.05). As shown in Figure 9, the reduction of the biofilm formation on Ag-doped materials was high and often reached 70%. A greater reduction was observed for alloys with a higher Ag concentration.

## 4. Conclusions

In this work, the preparation and characterization of new Ti-Ta-Ag bulk and porous alloys has been discussed. Such materials could find application in the production of hard tissue implants. The preparation procedures were related to the microstructure, mechanical, corrosion, and biological properties. Based on the presented investigations, the following conclusions can be drawn:Using mechanical alloying, and alloying elements such as Ta and Ag, enables achieving beta titanium, nanocrystalline powder alloys.The hot-pressing technique is beneficial in the development of high-density alloys. The new materials exhibit densities close to the theoretical value.The investigated alloying elements heavily influence the microstructure. Tantalum shows good potential in the grain size reduction in sintered materials.The new alloys show a good level of properties important from the biomaterial point of view (corrosion resistance, mechanical and wettability properties).The biological behavior, i.e., relative viability of the osteo- and fibro-blasts cells, as well as the antibacterial properties, were also high.

## Figures and Tables

**Figure 1 materials-15-04332-f001:**
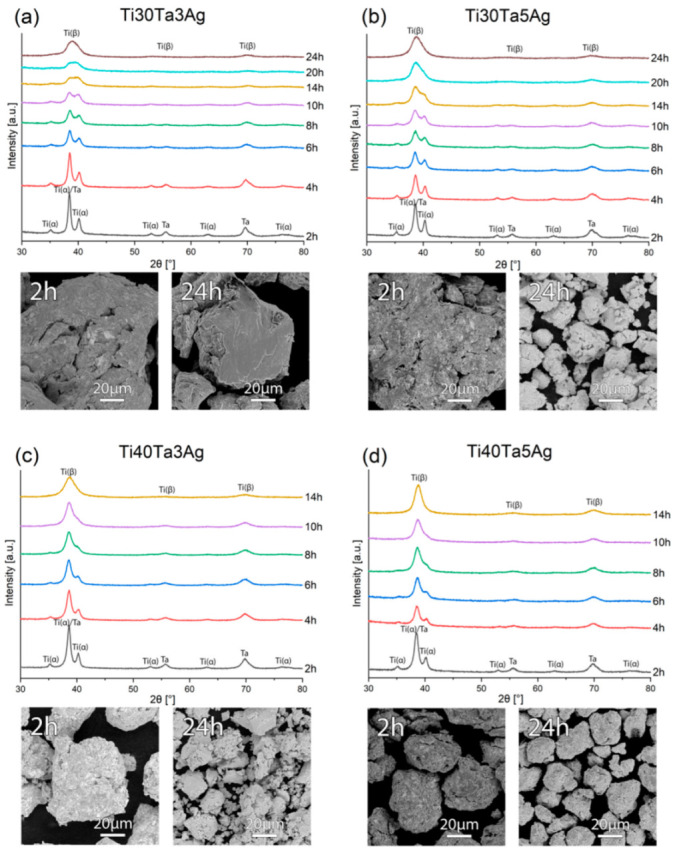
The XRD spectra and SEM micrographs of the (**a**) Ti30Ta3Ag, (**b**) Ti30Ta5Ag, (**c**) Ti40Ta3Ag, and (**d**) Ti40Ta5Ag powders at various milling times.

**Figure 2 materials-15-04332-f002:**
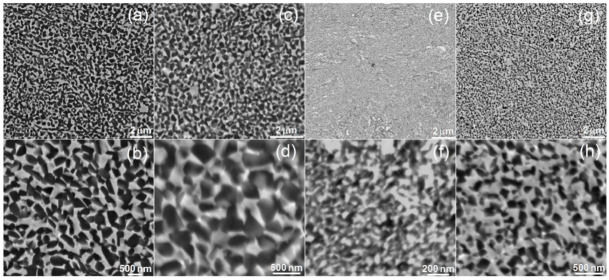
SEM micrographs of the (**a**,**b**) Ti30Ta3Ag, (**c**,**d**) Ti30Ta5Ag, (**e**,**f**) Ti40Ta3Ag, and (**g**,**h**) Ti40Ta5Ag microstructure—various magnifications.

**Figure 3 materials-15-04332-f003:**
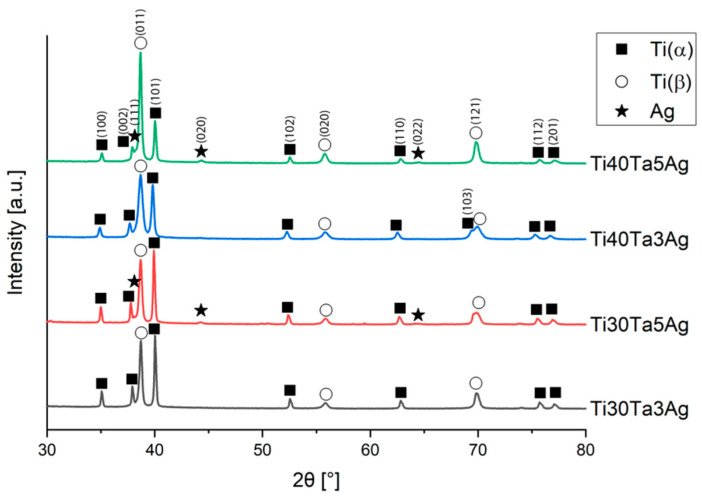
XRD spectra of the hot-pressed Ti-Ta-Ag alloys.

**Figure 4 materials-15-04332-f004:**
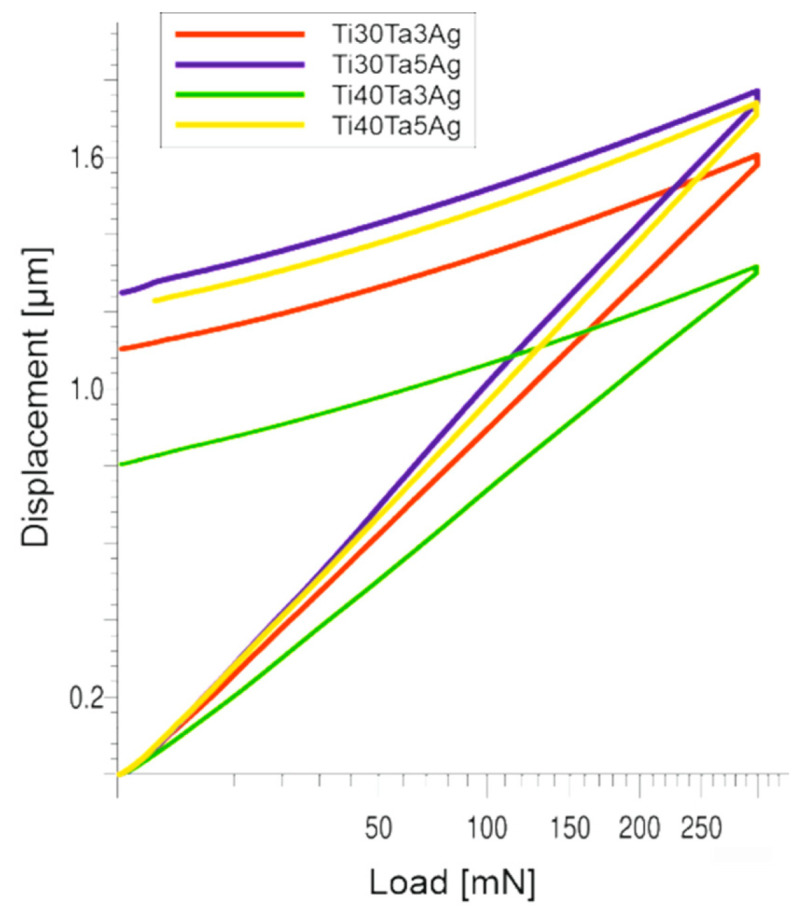
The nanoindentation (load-displacement) curves.

**Figure 5 materials-15-04332-f005:**
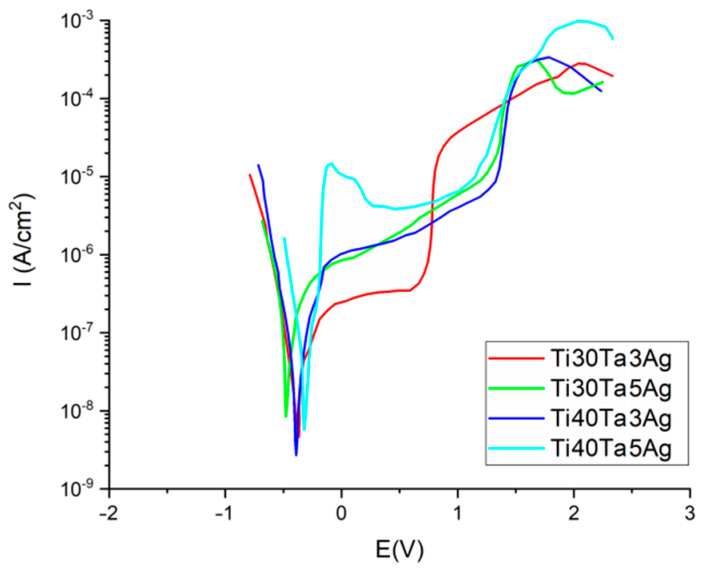
Potentiodynamic corrosion curves.

**Figure 6 materials-15-04332-f006:**
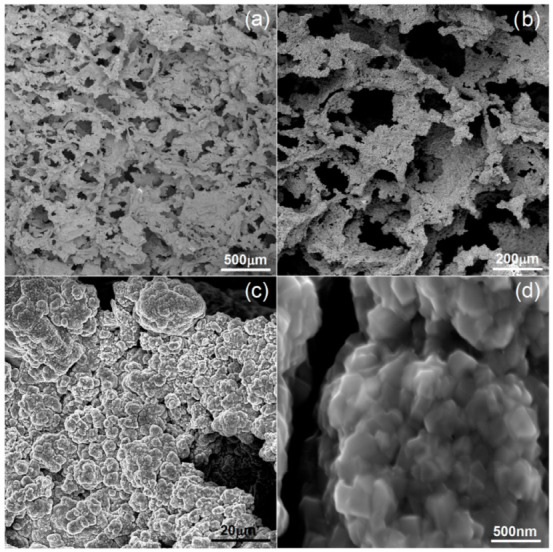
SEM micrographs of Ti40Ta3Ag foam: (**a**–**d**)—various magnifications.

**Figure 7 materials-15-04332-f007:**
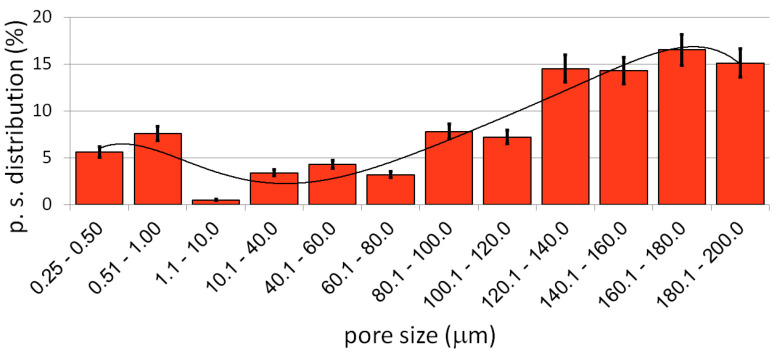
The pore size distribution of the Ti40Ta3Ag foam.

**Figure 8 materials-15-04332-f008:**
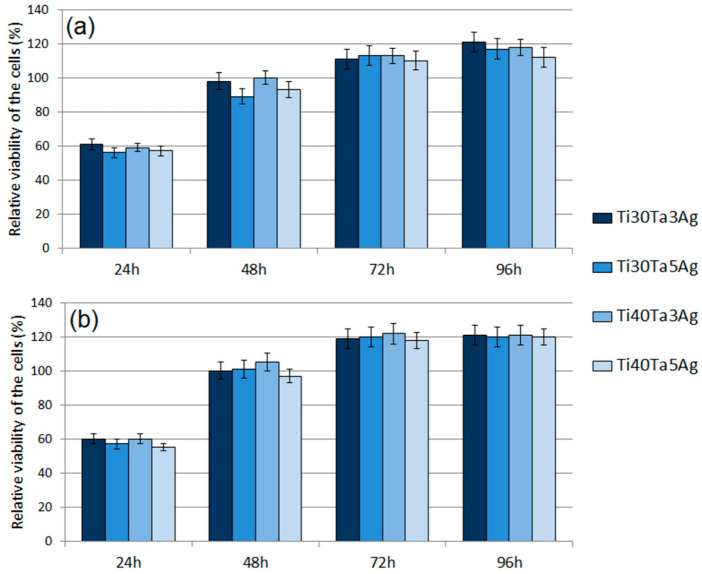
The results of the MTT assay performed at 24 h, 48 h, 72 h, and 96 h: (**a**) NHost cells and (**b**) HPLF cells.

**Figure 9 materials-15-04332-f009:**
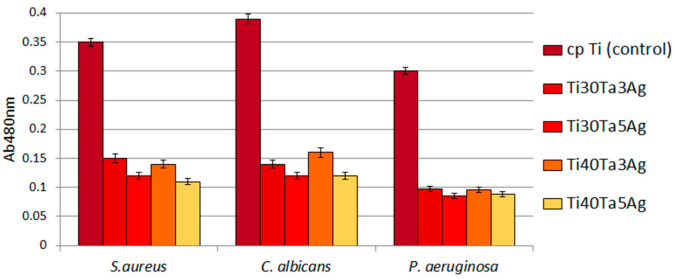
The reduced ability of tested microbes to form biofilm on Ti-Ta-Ag surface.

**Table 1 materials-15-04332-t001:** The crystallographic data of the powders after mechanical alloying and after hot-pressing; d—crystallite size; a, c—lattice constant; V—lattice volume; PA—phase amount; R_wp_—weighted pattern residual indicator; R_exp_—expected residual indicator; and S—goodness of fit.

Specimen	Powder Sample		Sintered Samples												
	d [nm]	Strain [%]	Ti (α)				Ti (β)			Additional Phase			R_wp_ [%]	R_exp_ [%]	S
										Ag					
			a [Å]	c [Å]	V [Å^3^]	PA [%]	a [Å]	V [Å^3^]	PA [%]	a [Å]	V [Å^3^]	PA [%]			
Ti30Ta3Ag	32	0.111	2.959	4.751	36.020	51.0	3.296	35.820	49.0	-	-	-	6.359	3.270	1.945
Ti30Ta5Ag	33	0.377	2.964	4.766	36.255	50.6	3.296	35.812	47.9	4.086	68.217	1.6	6.025	3.052	1.974
Ti40Ta3Ag	27	0.055	2.972	4.779	36.546	41.5	3.295	35.764	58.1	-	-	-	5.898	3.010	1.959
Ti40Ta5Ag	53	0.821	2.960	4.755	36.094	30.7	3.300	35.949	66.9	4.086	68.217	2.3	5.830	3.005	1.940

**Table 2 materials-15-04332-t002:** The mechanical properties of the Ti-Ta-Ag bulk alloys; the Vickers hardness (HV_0.3_), Martens hardness (HM), and Young’s modulus (E).

Material	HV_0.3_ ± σ	HM ± σ (N/mm^2^)	EIT ± σ (GPa)
Ti30Ta3Ag	628 ± 13	4250.8 ± 58	130.4 ± 1
Ti30Ta5Ag	497 ± 10	3553.5 ± 60	120.6 ± 2
Ti40Ta3Ag	971 ± 7	6095.0 ± 66	169.4 ± 4
Ti40Ta5Ag	572 ± 8	3941.0 ± 51	124.0 ± 1

**Table 3 materials-15-04332-t003:** Corrosion potential E_corr_ and corrosion current density I_corr_ for the Ti-Ta-Ag alloys.

Material	E_corr_	I_corr_
	(V)	(µA·cm^−2^)
Ti30Ta3Ag	−0.3691	0.14359
Ti30Ta5Ag	−0.46709	0.31849
Ti40Ta3Ag	−0.38724	0.092875
Ti40Ta5Ag	−0.32066	0.039116

**Table 4 materials-15-04332-t004:** The wettability data for the Ti-Ta-Ag hot-pressed alloys.

Measurement Name	Unit	Ti30Ta3Ag	Ti30Ta5Ag	Ti40Ta3Ag	Ti40Ta5Ag
Surface free energy	[mN/m]	35.08 (±9.2)	48.48 (±6.0)	33.02 (±7.1)	32.71 (±4.6)
Disperse	[mN/m]	10.93 (±3.7)	11.47 (±2.7)	15.30 (±3.7)	10.85 (±1.8)
Polar	[mN/m]	24.15 (±5.5)	37.01 (±3.4)	17.72 (±3.4)	21.86 (±2.7)
Water (Air) CA	[°]	67.16 (±2.7)	51.10 (±0. 8)	70.88 (±1.6)	70.13 (±1.5)
Glycerol (Air) CA	[°]	64.48 (±2.0)	50.51 (±2.4)	64.35 (±2.2)	67.05 (±0.8)

**Table 5 materials-15-04332-t005:** The mechanical properties of Ti-Ta-Ag foams.

Alloy	Compression Strength [MPa]	Elastic Moduli [GPa]
Ti30Ta3Ag	16.7 ± 0.5	0.91 ± 0.1
Ti30Ta5Ag	15.9 ± 0.6	0.89 ± 0.1
Ti40Ta3Ag	17.5 ± 0.5	1.27 ± 0.1
Ti40Ta5Ag	15.7 ± 0.7	0.98 ± 0.1

## Data Availability

Not applicable.

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
