# Peer review of "Preparation and Properties of Bulk and Porous Ti-Ta-Ag Biomedical Alloys"

_materials, 2022, doi:10.3390/ma15124332_

Round 1
Reviewer 1 Report
The manuscript presents experimental results on microstructure and properties of fabricated
Ti-Ta-Ag alloys by mechanical alloying, hot-pressing consolidation and sintering. The applied methodology and proposed processing route are relevant to make a conclusion about the application advantages of Ti-based alloys for hard tissue replacements. Undoubtedly, this paper will be attractive for readerships of Materials Journal. However, I cannot recommend this paper for publication in the current form due to the numerous drawbacks. My comments are given below:
1) Introduction section, line 38: «Such elements are: Zr, Nb and Ta, they belong to the non-toxic and non-allergic group of elements». This statement should be proved by citing the most relevant papers or book chapters, where the biocompatibility of Zr/Nb/Ta alloys was studied in detail.
2) Introduction section, line 41-43: « With the increasing concentration of Ta the elastic moduli decrease, strength increase [11-22] and corrosion resistance improve in compare with pure titanium, due to higher stability of Ta2O5 over TiO2 [23-28]». Please, check carefully the cited articles, because Ref. [11, 13] are not correct in this scope (Ta element was not chosen for alloying of some studied materials). Moreover, Ref. [23, 24, 28] do not contain any information about chemical or phase composition of oxide film (TiO2/Ta2O5), while only XPS or XRD method are able to reveal the structure of protective passivated layer. Also, in [24, 28] only CV, EIS data are presented that cannot prove/decline an issue of presence of different types of oxides.
3) In the end of introduction section the motivation of the research is required. Please, outline the aim of study. Are there any challenges in preparation of Ti-Ta alloys for biomedical application? What scientific problem do the authors try to solve?
4) Some important details are skipped in the Methodology section. Firstly, add the type of reference electrode (SCE, Ag/AgCl, etc.) used for corrosion resistance analysis. Secondly, XRD patterns were collected at unknown scan rate (?), scan step (?), applied voltage and current of experimental setup (?), using monochromator/Soller Slits or something else (?). Also, it is not enough to mentioned the label of utilized software. The procedure of Rietveld approach is complex, so the type of fitting function (Gaussian, Lorentzian, etc.) as well as the background subtraction are not trivial for XRD pattern analysis. Please, add the detailed procedure of Rietveld refinement used for your XRD patterns. Method of evaluation of crystallite size and microdeformation of the second kind is necessary to describe in Methodology section.
5) A lot of carelessness mistakes in the presentation of Fig. 1. The shown XRD patterns should be significantly revised due to the following. The authors should provide the clear deconvolution of diffraction peaks near 2Theta=38 deg. into (002)Ti, (110)Ta and (111)Ag for all (a-d) cases. Also, the peak (110) of Ti(beta) after 24 h (see Fig. 1a, b) has obvious shoulders that could belong to another phases. But, no peak splitting or peak deconvolution is given. I advise to present enlarged fragments of XRD patterns in the vicinity of 2Theta=34-45 deg. separately in order to prove the relevance of applied fitting procedure of individual XRD peaks. Please, add the scale bar for intensity (replace arb. unit. into counts/sec).
6) Based on the results of XRD analysis It’s not clear whether Ag dissolves into Ta (BCC) or Ti (HCP) matrix phase.
7) How did the authors evaluate the volume fraction of defined phases (table 1)? Please, add the description of the procedure of volume fraction estimation.
8) I suppose that strain of the second kind (Table 1, Ti30Ta3Ag, Ti40Ta3Ag) according to the Hall method is overestimated. How can the authors explain this huge lattice strain? Do the authors correctly evaluate the FWHM of peaks taking into account the Ka-dublet separation (Rechinger method), instrumental broadening.
9) Fig. 3, the authors describe the phase composition using «Ti(beta» and «Ti(alpha)» phases. Are they solid solution phases? If so, please, revise the labelling of peaks, for example «(Ti,Ta)-beta solid solution phase». Also, no main peak of (111)Ag is visible, however the volume fraction of Ag is given in Table 1. Instead, the Fig. 3 shows only (020) peak of Ag. Is it associated with the presence of texture? The presentation of Fig. 3 also includes the same drawbacks (see comment # 5). No clear peak deconvolution is presented. Therefore, for readers will not be convenient to check your results with other literature data.
10) Lines 222-224: «The solubility of Ag in titanium is relatively low and in room temperature reaches about 2-3%, so we could expect silver precipitates in the microstructure». Please, add the relevant Refs. with equilibrium phase diagrams of Ti/Ta-Ag.
11) Fig. 5, the potentiodynamic curves were measured versus reference electrode. What was the reference electrode?
12) In the end of manuscript, no discussion section is given. It’s not suitable for scientific papers. Any explanation of observed results is required.
Author Response
We would like to express our great appreciation to the Editor and respected Reviewers for examining the manuscript carefully and providing us valuable comments and constructive suggestions which are very important for the revision of the manuscript and improving the quality of the work. We have tried our best to revise our manuscript according to their suggestions.
Replies to Reviewer #1:
Reviewer #1:
Comment 1: Introduction section, line 38: «Such elements are: Zr, Nb and Ta, they belong to the non-toxic and non-allergic group of elements». This statement should be proved by citing the most relevant papers or book chapters, where the biocompatibility of Zr/Nb/Ta alloys was studied in detail.
Reply: improved – we add correct literature
Comment 2: Introduction section, line 41-43: « With the increasing concentration of Ta the elastic moduli decrease, strength increase [11-22] and corrosion resistance improve in compare with pure titanium, due to higher stability of Ta2O5 over TiO2 [23-28]». Please, check carefully the cited articles, because Ref. [11, 13] are not correct in this scope (Ta element was not chosen for alloying of some studied materials). Moreover, Ref. [23, 24, 28] do not contain any information about chemical or phase composition of oxide film (TiO2/Ta2O5), while only XPS or XRD method are able to reveal the structure of protective passivated layer. Also, in [24, 28] only CV, EIS data are presented that cannot prove/decline an issue of presence of different types of oxides.
Reply: improved – we add correct literature
Comment 3: In the end of introduction section the motivation of the research is required. Please, outline the aim of study. Are there any challenges in preparation of Ti-Ta alloys for biomedical application? What scientific problem do the authors try to solve?
Reply: improved – we add more comments in introduction supporting the motivation
Comment 4: Some important details are skipped in the Methodology section. Firstly, add the type of reference electrode (SCE, Ag/AgCl, etc.) used for corrosion resistance analysis. Secondly, XRD patterns were collected at unknown scan rate (?), scan step (?), applied voltage and current of experimental setup (?), using monochromator/Soller Slits or something else (?). Also, it is not enough to mentioned the label of utilized software. The procedure of Rietveld approach is complex, so the type of fitting function (Gaussian, Lorentzian, etc.) as well as the background subtraction are not trivial for XRD pattern analysis. Please, add the detailed procedure of Rietveld refinement used for your XRD patterns. Method of evaluation of crystallite size and microdeformation of the second kind is necessary to describe in Methodology section.
Reply: improved
Comment 5: A lot of carelessness mistakes in the presentation of Fig. 1. The shown XRD patterns should be significantly revised due to the following. The authors should provide the clear deconvolution of diffraction peaks near 2Theta=38 deg. into (002)Ti, (110)Ta and (111)Ag for all (a-d) cases. Also, the peak (110) of Ti(beta) after 24 h (see Fig. 1a, b) has obvious shoulders that could belong to another phases. But, no peak splitting or peak deconvolution is given. I advise to present enlarged fragments of XRD patterns in the vicinity of 2Theta=34-45 deg. separately in order to prove the relevance of applied fitting procedure of individual XRD peaks. Please, add the scale bar for intensity (replace arb. unit. into counts/sec).
Reply: After consultation our opinion is that there is no shoulders for the peak (110) of Ti(beta) after 24 h. The significant broadening of the peak is a typical effect after mechanical alloying. We see no reason to give enlarged fragment of XRD. We declare that the analysis was carried out correctly. While searching through the other articles, most majority have arbitrary units.
Comment 6: Based on the results of XRD analysis It’s not clear whether Ag dissolves into Ta (BCC) or Ti (HCP) matrix phase.
Reply: According to the phase diagrams of Ti-Ag and Ta-Ag, silver dissovles into titanium. Tantalum do not make such easy solutions with Ag.
Comment 7: How did the authors evaluate the volume fraction of defined phases (table 1)? Please, add the description of the procedure of volume fraction estimation.
Reply: Phases volume were evaluated with a standardless X-ray, quantitive phase analysis.
Comment 8: I suppose that strain of the second kind (Table 1, Ti30Ta3Ag, Ti40Ta3Ag) according to the Hall method is overestimated. How can the authors explain this huge lattice strain? Do the authors correctly evaluate the FWHM of peaks taking into account the Ka-dublet separation (Rechinger method), instrumental broadening.
Reply: After closer revision of the obtained results, we corrected the values in a table for all powdery samples. Ka-dublet separation and instrumental broadening were considered in FWHM evaluations.
Comment 9: Fig. 3, the authors describe the phase composition using «Ti(beta» and «Ti(alpha)» phases. Are they solid solution phases? If so, please, revise the labelling of peaks, for example «(Ti,Ta)-beta solid solution phase». Also, no main peak of (111)Ag is visible, however the volume fraction of Ag is given in Table 1. Instead, the Fig. 3 shows only (020) peak of Ag. Is it associated with the presence of texture? The presentation of Fig. 3 also includes the same drawbacks (see comment # 5). No clear peak deconvolution is presented. Therefore, for readers will not be convenient to check your results with other literature data.
Reply: In both Ti beta and Ti alpha tantalum is present, but in much lower concentration in alpha as in beta. The main peak from Ag is not presented, because we wanted to make the graph more readable. Missing peak has been added
Comment 10: Lines 222-224: «The solubility of Ag in titanium is relatively low and in room temperature reaches about 2-3%, so we could expect silver precipitates in the microstructure». Please, add the relevant Refs. with equilibrium phase diagrams of Ti/Ta-Ag.
Reply: improved
Comment 11: Fig. 5, the potentiodynamic curves were measured versus reference electrode. What was the reference electrode?
Reply: The potentiodynamic curves were measured vs. OCP. This and type of reference electrode are describe in MM part.
Comment 12: In the end of manuscript, no discussion section is given. It’s not suitable for scientific papers. Any explanation of observed results is required.
Reply: improved
Reviewer 2 Report
The paper by G. Adamek et al is devoted to the development of new alloy powders in Ti-Ta-Ag system using mechanical alloying. The authors used MA powders to produce bulk and porous materials and comprehensively studied their structure, mechanical and biological properties. The experiments are well designed, and the obtained results are promising for the use of designed alloys in bio-application. However, the manuscript should be revised to be published in Materials journal.
- The abbreviation MMT should be deciphered in the abstract
- The phrase “high corrosion resistance” is duplicated in the first sentence.
- In the introduction part (page 1, line 39) “They should stabilize beta structure, which shows better Young’s moduli”. The authors should indicate the structure of which phase they have in mind as this is not clear from the context.
- At the end of the introduction part is written “The main goal of this study is to investigate the effect of different content of (30, 40%) Ta with small amounts (3, 5%) of Ag in titanium alloys, as well as preparation parameters on the microstructure, mechanical properties, corrosion resistance and biological behavior”. While at the Experimental part is written “Based on our previous initial studies and literature, we decided to investigate the following chemical compositions: TixTayAg, where x = 30, 40 wt% and y = 3, 5 wt%”. Readers may be confused if x refers to Ti or Ta, as well as if y refers to Ta or Ag. Please, make this sentence clearer.
- The materials of the balls used for mechanical alloying should be mentioned.
- At the page 4 the authors discussed the influence of Ag addition on the crystallite size of the alloy powders. The authors made the conclusion that that there is an increase of crystal size with increasing Ag content, independently of Ta amount. Can the authors discuss a possible reason of the observed behaviour?
- Table 1 should be corrected to be readable. The precision of measurements of lattice parameters is very high (6 digits after the comma!). Do the authors use a standard powder to perform XRD?
- The authors claim that “Another way to improve the elastic moduli is to prepare porous materials, which is the next step of this study”. However, the part of the manuscript devoted to the preparation of porous materials based on alloy powders does not contain results on Young’s moduli.
- Page 9, line 227 “The total porosity is approx. 70%, which nicely correspond with theoretical calculations and comparison with other authors study [40-43]”. It is not clear which theoretical calculations the authors have in mind.
- The authors should be precise in their conclusion. For example, the statement “the investigated alloying elements strongly influence the microstructure. Tantalum show good potential in grain size reduction” can refer to powders, bulk or porous materials.
Author Response
We would like to express our great appreciation to the Editor and respected Reviewers for examining the manuscript carefully and providing us valuable comments and constructive suggestions which are very important for the revision of the manuscript and improving the quality of the work. We have tried our best to revise our manuscript according to their suggestions.
Replies to Reviewer #2:
Reviewer #2:
Comment 1: The abbreviation MMT should be deciphered in the abstract
Reply: The 3-(4,5-dimethylthiazol-2-yl)-2,5-diphenyl-2H-tetrazolium bromide (MTT) assay has become the gold standard for determination of cell viability and proliferation since its development by Mosmann in the 1980′s. In literature the “MTT assay” is commonly used as a name of the test and in our opinion there is no need to defined it here.
Comment 2: The phrase “high corrosion resistance” is duplicated in the first sentence.
Reply: improved
Comment 3: In the introduction part (page 1, line 39) “They should stabilize beta structure, which shows better Young’s moduli”. The authors should indicate the structure of which phase they have in mind as this is not clear from the context.
Reply: improved
Comment 4: At the end of the introduction part is written “The main goal of this study is to investigate the effect of different content of (30, 40%) Ta with small amounts (3, 5%) of Ag in titanium alloys, as well as preparation parameters on the microstructure, mechanical properties, corrosion resistance and biological behavior”. While at the Experimental part is written “Based on our previous initial studies and literature, we decided to investigate the following chemical compositions: TixTayAg, where x = 30, 40 wt% and y = 3, 5 wt%”. Readers may be confused if x refers to Ti or Ta, as well as if y refers to Ta or Ag. Please, make this sentence clearer.
Reply: improved
Comment 5: The materials of the balls used for mechanical alloying should be mentioned.
Reply: improved
Comment 6: At the page 4 the authors discussed the influence of Ag addition on the crystallite size of the alloy powders. The authors made the conclusion that that there is an increase of crystal size with increasing Ag content, independently of Ta amount. Can the authors discuss a possible reason of the observed behaviour?
Reply: After a closer revision of the results, the statement was removed.
Comment 7: Table 1 should be corrected to be readable. The precision of measurements of lattice parameters is very high (6 digits after the comma!). Do the authors use a standard powder to perform XRD?
Reply: Table have been corrected as well as the presented values. Yes, standard powder.
Comment 8: The authors claim that “Another way to improve the elastic moduli is to prepare porous materials, which is the next step of this study”. However, the part of the manuscript devoted to the preparation of porous materials based on alloy powders does not contain results on Young’s moduli.
Reply: improved, we add new results
Comment 9: Page 9, line 227 “The total porosity is approx. 70%, which nicely correspond with theoretical calculations and comparison with other authors study [40-43]”. It is not clear which theoretical calculations the authors have in mind.
Reply: improved – this part of text was modified
Comment 10: The authors should be precise in their conclusion. For example, the statement “the investigated alloying elements strongly influence the microstructure. Tantalum show good potential in grain size reduction” can refer to powders, bulk or porous materials.
Reply: improved
Reviewer 3 Report
The manuscript entitled “Preparation and properties of bulk and porous Ti-Ta-Ag biomedical alloys” by Grzegorz Adamek al. is a research article that tackles the porosity, morphology-microstructure and mechanical properties of different Ti-based alloys. Here they compare samples ranging from bulk to different representative micro-grained specimens prepared by mechanical alloying. The article discusses the crystal structure, microstructure and some mechanical properties such as load-displacement (strain-stress curves), material hardness (Vickers and Martens hardness along with the Young’s modulus) with respect to the chemical composition and considering their biocompatibility and antibacterial properties of such Ti-based alloys.
This is an interesting work that gathers several techniques to composes a comprehensive study with important applications in medicine as hard-tissue implants. Unfortunately, the manuscript lacks consistency and structure and sometimes is hard to follow because is read like a patchwork of information, but without a clear line of argument. Therefore, I cannot recommend this manuscript for publication in its current form. The language needs to be improved, and typos checked more carefully. Hence, I recommend to reconsider the manuscript only after major adjustments (in the whole manuscript).
Points to be addressed (I just list sme of them):
- The manuscript needs an important English revision:
- Rewrite l.14 “The paper presents results of preparation of bulk and porous Ti-Ta-Ag alloys”.
- 15 remove comma “the second,…”
- 16 modify “,result”, by ““,which result+s”.
- 12 replace “The new materials…” by “These new materials…”
- 21 “application+s”
- 22 “implant+s”
- 26 include “… alloys are the most widely …”
- 29 lacks a subject “for permanent implants it shows …”
- 35 “…enhance+s body fluid….”
- 36 add a which “portation, which results…”, and “Newly design+ed …”
- 42 “increase+s, (while) strength increase+s, … improve+s in )comparison) …”
- 47 “improve+s”
- 48: shows an extra comma
- 58 “… content+s”
- 61: “(In this work) We investigate both, bulk and porous samples.”
- 145: “pro(p)erties”
- 156: replace “un(-)bound” for “unbound”
- 160: could authors check spelling of “formazan(e)”
- 163 “formazan suspended in medium…”
- 178: “results confirm that(:) Alloys with” Add “:”
- 213-214: Please, check English of “Figure 3. shows the XRD spectra of hot-pressed alloys. (Every) alloy exhibited a two-phase structure of Ti- and Ti- which confirm(ed) microscopic results (observations)”
- Initials/acronyms not defined:
- 21: MTT maybe means “mean transit time” and is a method to evaluate the hazard assessment of nanoparticles. The first time an Initial is used, must be defined. In addition, take into account that this is not a biomedical journal, so the audience need to be informed about such specific terminology.
- 72: “XRD” and “SEM” initials need to be defined.
- 106: “SE”, “BSE” and “EDS” initials need to be defined.
- 145: “ATCC”
- 147 “TSB”
- the manuscript lacks consistency and is hard to follow:
Needs clarification or rephrasing:
- 28-32:“Moreover, in a group of materials used for permanent implants (it) shows the lowest Young modulus and density (density is also reduced ?), but still, the elastic modulus is high (65-110 GPa) in comparison with a bone (10-30 GPa) [1-4], (which) could results (extra “s”) in bone resorption (which) in the worst scenario may lead to implant loosening or bone fracture.” Among other issues, authors employ 2 “which” in the same subordinate sentence.
- 32-33:“There are two ways to reduce the elastic moduli: (firstly) the appropriate combination of chemical composition and(, the second and) more effective (one,)(:) preparation of (a) special form of the material with high porosity called metallic foams”.
- 38.39: “Such elements are: Zr, Nb and Ta(, t) . They belong to (the)(a) non-toxic and non-allergic group of elements”
- 44-45: “Silver is (an element) well-known for its antibacterial properties.”
- 45-46: “ The (a)ddition of few (%Ag) into the alloy, helps to keep the area of implantation free of bacteria and prevents to form biofilm [29-31].” Authors must clarify if they mean weight percentage (wt.%) of Ag, or in the stoichiometric composition of the Ti-Ta-Ag alloys, or what do they mean ? Have a look in line 59 and 67 !
- 58-61: Usually, I found that compositional stoichiometry is show as follows: TiXTayAgz or Ti100-x-yTaxAgy, but never TixTayAg. Please, revise this in the while manuscript. This could lead to misunderstood. In addition, please, check if the percentage elements is weight percentage or what and follow the same criteria in the whole manuscript for consistency.
- 59-60: authors talk about “preparation parameters on the microstructure”, please, could the authors include a couple of them into brackets for clarity.
- 130-143: To grab the attention of a broad audience and for non-experts in the field, which is my case, I believe that the manuscript would increase their scope if the authors could broadly explain in what consist the MTT assay, and the particular importance of the MTT assay (including RVC value) in this study. Underlying the importance of MTT assay (including RVC value) not only in the Methods sections, but also somewhere in the manuscript.
- 138: According to literature “pure microcrystalline cp-titanium” means that it is above > 98% of Ti (I do not know if wt.%), whereas authors stated it is of the 100%. Could authors clarify this point ? Besides, this 2% of spurious elements could introduce “spurious” effects misleading the final conclusions od the work ? Have the authors checked the purity of the “pure microcrystalline cp-titanium”?
- 165-66: Please, rephrase ” (Statistical analysis) was performed using (Statistical calculations) were performed with the SigmaStat package, Version 2.0 (SPSS, Chicago, IL) [ref]” and include a reference.
- 171:authors states that samples are nano-grained “In this paper the ultrafine- / nano-grained Ti-Ta-Ag alloys were investigated”, whereas their observations stated that grains are of micron size (l.198). Please, correct this.
- 180-181:Please, could authors rephrase this “In all cases the amount of Ag was low enough to peaks correspond to the element disappearing…”, for clarity.
- 207:please, rephrase this sentence “Using hot-pressing and optimiz(e) sintering parameters (there) is possible to achieve high”.
- 217-218: include “,and compiled in table(1?).”, at the end of the sentence “The crystallographic data have been calculated using Rietveld refinement using High-Score Plus software.”
- 249: Please, rephrase this “The impact of Ta is strongly highlighted than Ag.”
- 270-271: “In Figure 6a) we can observe a general view and higher magnification in Fig. 6b)”. Authors must rephrase this.
Assertions need to be referenced or explained:
- 39-41: “They should stabilize beta structure, which shows better Young’s moduli[ref]. Tantalum is considered to be one of the strongest beta stabilizers [ref].”
- 49: “…relatively high melting point [ref].” Needs to include Ti melting point (ca. 1940 K ) along with the reference.
- 65-66: “Based on our previous initial studies and literature [ref.], we decided to investigate the following chemical compositions”. Authors need to include references and justify in a brief sentence their chemical composition selection.
- 70-72: Authors need to explain and/or include a reference of the criteria used to stop the milling time. In addition, please, include a sentence to advise the audience that XRD and SEM experimental measurements are explained in the following subsections (below).
- 100-103: Authors must include the Cuα wavelength , and the HighScore Plus Software must be referenced.
- L:210-211: Need for a reference “The pores could negatively influence mechanical parameters, so in developing bulk materials it is important to reduce it.”
- Subsections are not numbered in the Materials and Methods section, i.e., ranging from 2.1 to 2.8, if I am right.
- In line 186: Authors must explain how the authors calculated the “average” particle size, indicating the expressions/software used and referencing it. In addition, they must show the uncertainty of the measurement and how it was obtained.
- In lines 196-197, authors must explained why T-alpha areas are dark, whereas T-beta+T-alpha regions are bright.
- In lines 198-201, authors must include the method used (and reference it) to calculate grain-size and particle size, along with its uncertainties.
- For Clarity, authors could include another table comparing chemical composition, milling time, crystal-size and grain-size. This will simplify the line of reasoning explained in lines 194-203. Authors discuss about milling time, but is not clearly included in the manuscript.
- In fig.2, authors can add the measured grain-size in each panel for all the samples.
- All Figure captions must be self-consistent and self-explanatory for clarity to the audience.
- L208-29: “The porosity of prepared alloys was as follows: 0.03, 0.03, 0.05, 0.07% for Ti30Ta3Ag, Ti30Ta5Ag, Ti40Ta3Ag and Ti40Ta5Ag, respectively”. As a non-expert in the field, I would appreciate if authors could explain how porosity is measured and what this numbers account for, e.g., accompanied by the formula used in the calculations and a reference.
- 213-218: Authors claim there is a solid-solid phase transition, but there is NO trace of refinement shown in the Figures. In fact, in table 1 authors show the crystallographic parameters of the refinement, but there is NO TRACE of the space group or crystal structure used in the refinements. Authors can also show a reference with the space group used in the literature. This lacks consistency in the findings shown in table 1.
- Authors must improve the performance of table 1 for clarity and visual aspect of the information compiled there, it is quite cramped/without space and thus hard to follow.
- In table 1, please check the strain value/units, it is shown in “o/ooo”, instead of “%” (e.g., fullprof suite software).
- 227-228: Please include the values in the literature to be able for the comparison (not only the reference). There is no reason for speculation.
- 232: “From the data we can clearly see that increasing …” What data ? Do authors mean table 2 ?
- 238: “The reason for such results obtained in this work could be very fine microstructure as well as solid solution strengthening.” I do not follow authors, what do authors mean? Please, authors could rewrite this for clarity.
- L239-240:” Another way to improve the (elastic moduli) is to prepare porous materials, which is the next step of this study. ” With elastic modulus, authors refer to Young modulus? In addition, I would appreciate if authors could clarify this point. I thought authors were already studying porous materials, but the last sentence infers the other way around. This kind of things blurs the line of argument of the manuscript. Please, revise this.
- Please, enlarge fig.4 and use a different “line+symbol” for each composition to make it easy to follow the figure from visual inspection. This is a general advice for all the figures with experimental data. In addition, this enhance the performance in black & white prints.
- The explanation of figure 4 must be improved in the text.
- Please, homogenize all the axis labels in all the figures. In particular, axis and number labels are quite small in Fig4.
- To reach a wide audience, I would appreciate if authors could clarify the meaning of “Ecorr” and “Icorr” parameters, along with the mathematical expression (if exists) and a brief explanation of their fundamentals, just to understand the numbers un the text (in line 246), while putting into context the importance of data gathered in table 3.
- As well as for the corrosion properties, the wetting properties need to be improved. It is true that I am not in the field, but I found it difficult to follow and without a clear line of argument. Authors must improve this part of the manuscript to gain consistency.
- Please, could you explain the meaning of y-axis label in Figure 9, and expand explanation of fig.9 in the text.
Author Response
We would like to express our great appreciation to the Editor and respected Reviewers for examining the manuscript carefully and providing us valuable comments and constructive suggestions which are very important for the revision of the manuscript and improving the quality of the work. We have tried our best to revise our manuscript according to their suggestions.
Replies to Reviewer #3:
Reviewer #3:
Comment 1: The manuscript needs an important English revision:
- Rewrite l.14 “The paper presents results of preparation of bulk and porous Ti-Ta-Ag alloys”.
- 15 remove comma “the second,…”
- 16 modify “,result”, by ““,which result+s”.
- 12 replace “The new materials…” by “These new materials…”
- 21 “application+s”
- 22 “implant+s”
- 26 include “… alloys are the most widely …”
- 29 lacks a subject “for permanent implants it shows …”
- 35 “…enhance+s body fluid….”
- 36 add a which “portation, which results…”, and “Newly design+ed …”
- 42 “increase+s, (while) strength increase+s, … improve+s in )comparison) …”
- 47 “improve+s”
- 48: shows an extra comma
- 58 “… content+s”
- 61: “(In this work) We investigate both, bulk and porous samples.”
- 145: “pro(p)erties”
- 156: replace “un(-)bound” for “unbound”
- 160: could authors check spelling of “formazan(e)”
- 163 “formazan suspended in medium…”
- 178: “results confirm that(:) Alloys with” Add “:”
- 213-214: Please, check English of “Figure 3. shows the XRD spectra of hot-pressed alloys. (Every) alloy exhibited a two-phase structure of Ti-b and Ti-a which confirm(ed) microscopic results (observations)”
Reply: The text was improved according to reviewer comments and English language was corrected by professional translator
Comment 2: Initials/acronyms not defined:
- 21: MTT maybe means “mean transit time” and is a method to evaluate the hazard assessment of nanoparticles. The first time an Initial is used, must be defined. In addition, take into account that this is not a biomedical journal, so the audience need to be informed about such specific terminology.
- 72: “XRD” and “SEM” initials need to be defined.
- 106: “SE”, “BSE” and “EDS” initials need to be defined.
- 145: “ATCC”
- 147 “TSB”
Reply: We defined the acronyms, however, without MTT assay. The MTT acronym comes from 3-(4,5-Dimethylthiazol-2-yl)-2,5-Diphenyltetrazolium Bromide. In literature the “MTT assay” is commonly used as a name of the test and in our opinion there is no need to defined it here.
Comment 3: 28-32:“Moreover, in a group of materials used for permanent implants (it) shows the lowest Young modulus and density (density is also reduced ?), but still, the elastic modulus is high (65-110 GPa) in comparison with a bone (10-30 GPa) [1-4], (which) could results (extra “s”) in bone resorption (which) in the worst scenario may lead to implant loosening or bone fracture.” Among other issues, authors employ 2 “which” in the same subordinate sentence.
Reply: improved
Comment 4: 32-33:“There are two ways to reduce the elastic moduli: (firstly) the appropriate combination of chemical composition and(, the second and) more effective (one,)(:) preparation of (a) special form of the material with high porosity called metallic foams”.
Reply: improved
Comment 5: 38.39: “Such elements are: Zr, Nb and Ta(, t) . They belong to (the)(a) non-toxic and non-allergic group of elements”
Reply: improved
Comment 6: 44-45: “Silver is (an element) well-known for its antibacterial properties.”
Reply: improved
Comment 7: 45-46: “ The (a)ddition of few (%Ag) into the alloy, helps to keep the area of implantation free of bacteria and prevents to form biofilm [29-31].” Authors must clarify if they mean weight percentage (wt.%) of Ag, or in the stoichiometric composition of the Ti-Ta-Ag alloys, or what do they mean ? Have a look in line 59 and 67 !
Reply: improved
Comment 8: 58-61: Usually, I found that compositional stoichiometry is show as follows: TiXTayAgz or Ti100-x-yTaxAgy, but never TixTayAg. Please, revise this in the while manuscript. This could lead to misunderstood. In addition, please, check if the percentage elements is weight percentage or what and follow the same criteria in the whole manuscript for consistency.
Reply: The Ti-xTa-yAg stoichiometry writing is often used. The most common Ti6Al4V alloy consist of 90wt% of Ti, 6wt% of Al and 4wt% of V.
Comment 9: 59-60: authors talk about “preparation parameters on the microstructure”, please, could the authors include a couple of them into brackets for clarity.
Reply: improved
Comment 10: 130-143: To grab the attention of a broad audience and for non-experts in the field, which is my case, I believe that the manuscript would increase their scope if the authors could broadly explain in what consist the MTT assay, and the particular importance of the MTT assay (including RVC value) in this study. Underlying the importance of MTT assay (including RVC value) not only in the Methods sections, but also somewhere in the manuscript.
Reply: The MTT assay is used to measure cellular metabolic activity as an indicator of cell viability, proliferation and cytotoxicity.
Comment 11: 138: According to literature “pure microcrystalline cp-titanium” means that it is above > 98% of Ti (I do not know if wt.%), whereas authors stated it is of the 100%. Could authors clarify this point ? Besides, this 2% of spurious elements could introduce “spurious” effects misleading the final conclusions od the work ? Have the authors checked the purity of the “pure microcrystalline cp-titanium”?
Reply: “The reference medium were prepared using pure bulk microcrystalline cp-titanium samples and it was represented by 100%.” The meaning of the sentence is that the cp-titanium samples were used to prepared the reference medium and the value in fig. 8 for the reference medium was represented by 100%.
Comment 12: 165-66: Please, rephrase ” (Statistical analysis) was performed using (Statistical calculations) were performed with the SigmaStat package, Version 2.0 (SPSS, Chicago, IL) [ref]” and include a reference.
Reply: improved
Comment 13: 171:authors states that samples are nano-grained “In this paper the ultrafine- / nano-grained Ti-Ta-Ag alloys were investigated”, whereas their observations stated that grains are of micron size (l.198). Please, correct this.
Reply: improved
Comment 14: 180-181:Please, could authors rephrase this “In all cases the amount of Ag was low enough to peaks correspond to the element disappearing…”, for clarity.
Reply: improved
Comment 15: 207:please, rephrase this sentence “Using hot-pressing and optimiz(e) sintering parameters (there) is possible to achieve high”.
Reply: improved
Comment 16: 217-218: include “,and compiled in table(1?).”, at the end of the sentence “The crystallographic data have been calculated using Rietveld refinement using High-Score Plus software.”
Reply: improved
Comment 17: 249: Please, rephrase this “The impact of Ta is strongly highlighted than Ag.”
Reply: improved
Comment 18: 270-271: “In Figure 6a) we can observe a general view and higher magnification in Fig. 6b)”. Authors must rephrase this.
Reply: improved
Comment 19: 39-41: “They should stabilize beta structure, which shows better Young’s moduli[ref]. Tantalum is considered to be one of the strongest beta stabilizers [ref].”
Reply: improved
Comment 20: 49: “…relatively high melting point [ref].” Needs to include Ti melting point (ca. 1940 K ) along with the reference.
Reply: improved
Comment 21: 65-66: “Based on our previous initial studies and literature [ref.], we decided to investigate the following chemical compositions”. Authors need to include references and justify in a brief sentence their chemical composition selection.
Reply: the literature were cite in introduction part – in our opinion there is no need to repeat it here
Comment 22: 70-72: Authors need to explain and/or include a reference of the criteria used to stop the milling time. In addition, please, include a sentence to advise the audience that XRD and SEM experimental measurements are explained in the following subsections (below).
Reply: improved
Comment 23: 100-103: Authors must include the Cuα wavelength, and the HighScore Plus Software must be referenced.
Reply: improved
Comment 24: L:210-211: Need for a reference “The pores could negatively influence mechanical parameters, so in developing bulk materials it is important to reduce it.”
Reply: In our opinion this is too basic knowledge to support it by reference.
Comment 24: 4.Subsections are not numbered in the Materials and Methods section, i.e., ranging from 2.1 to 2.8, if I am right.
Reply: improved
Comment 25: 5. In line 186: Authors must explain how the authors calculated the “average” particle size, indicating the expressions/software used and referencing it. In addition, they must show the uncertainty of the measurement and how it was obtained.
Reply: improved
Comment 26: 6. In lines 196-197, authors must explained why T-alpha areas are dark, whereas T-beta+T-alpha regions are bright.
Reply: In our opinion this is too basic knowledge to support it by reference. When we use BSE detector the number of backscattered electrons (BSE) is proportional to the mean atomic number of the sample. Thus, a "brighter" BSE intensity correlates with greater average Z in the sample, and "dark" areas have lower average Z.
Comment 27: 7. In lines 198-201, authors must include the method used (and reference it) to calculate grain-size and particle size, along with its uncertainties.
Reply: improved – we add information in MM part
Comment 28: 8. For Clarity, authors could include another table comparing chemical composition, milling time, crystal-size and grain-size. This will simplify the line of reasoning explained in lines 194-203. Authors discuss about milling time, but is not clearly included in the manuscript.
Reply: in our opinion another table will be unnecessary copying information from the text. After consultation we decided to not include extra table.
Comment 29: 9. In fig.2, authors can add the measured grain-size in each panel for all the samples.
Reply: We believe that present form of Fig. 2. is optimal and correspond to scientific papers style
Comment 30: 10. All Figure captions must be self-consistent and self-explanatory for clarity to the audience.
Reply: in our opinion the captions are communicable
Comment 31: 11. L208-29: “The porosity of prepared alloys was as follows: 0.03, 0.03, 0.05, 0.07% for Ti30Ta3Ag, Ti30Ta5Ag, Ti40Ta3Ag and Ti40Ta5Ag, respectively”. As a non-expert in the field, I would appreciate if authors could explain how porosity is measured and what this numbers account for, e.g., accompanied by the formula used in the calculations and a reference.
Reply: we add information about porosity measurements in MM part
Comment 32: 12. 213-218: Authors claim there is a solid-solid phase transition, but there is NO trace of refinement shown in the Figures. In fact, in table 1 authors show the crystallographic parameters of the refinement, but there is NO TRACE of the space group or crystal structure used in the refinements. Authors can also show a reference with the space group used in the literature. This lacks consistency in the findings shown in table 1.
Reply: improved
Comment 33: 13. Authors must improve the performance of table 1 for clarity and visual aspect of the information compiled there, it is quite cramped/without space and thus hard to follow.
Reply: improved
Comment 34: 14. In table 1, please check the strain value/units, it is shown in “o/ooo”, instead of “%” (e.g., fullprof suite software).
Reply: HighScore Plus presents these values in %
Comment 35: 15. 227-228: Please include the values in the literature to be able for the comparison (not only the reference). There is no reason for speculation.
Reply: improved
Comment 36: 16. 232: “From the data we can clearly see that increasing …” What data ? Do authors mean table 2 ?
Reply: Yes, Table 2. The information about it is two lines above
Comment 37: 17. 238: “The reason for such results obtained in this work could be very fine microstructure as well as solid solution strengthening.” I do not follow authors, what do authors mean? Please, authors could rewrite this for clarity.
Reply: improved
Comment 38: 18. L239-240:” Another way to improve the (elastic moduli) is to prepare porous materials, which is the next step of this study. ” With elastic modulus, authors refer to Young modulus? In addition, I would appreciate if authors could clarify this point. I thought authors were already studying porous materials, but the last sentence infers the other way around. This kind of things blurs the line of argument of the manuscript. Please, revise this.
Reply: improved
Comment 39: 19. Please, enlarge fig.4 and use a different “line+symbol” for each composition to make it easy to follow the figure from visual inspection. This is a general advice for all the figures with experimental data. In addition, this enhance the performance in black & white prints.
Reply: we enlarge the figure. We agree that in black & white prints it could be better to have “line+symbol”, however, nowadays more popular is to read papers on electronic devices and in our opinion color lines are more attractive for the readers.
Comment 40: 20. The explanation of figure 4 must be improved in the text.
Reply: improved
Comment 41: 21. Please, homogenize all the axis labels in all the figures. In particular, axis and number labels are quite small in Fig4.
Reply: improved
Comment 42: 22. To reach a wide audience, I would appreciate if authors could clarify the meaning of “Ecorr” and “Icorr” parameters, along with the mathematical expression (if exists) and a brief explanation of their fundamentals, just to understand the numbers un the text (in line 246), while putting into context the importance of data gathered in table 3.
Reply: improved
Comment 43: 23. As well as for the corrosion properties, the wetting properties need to be improved. It is true that I am not in the field, but I found it difficult to follow and without a clear line of argument. Authors must improve this part of the manuscript to gain consistency.
Reply: improved
Comment 44: 24. Please, could you explain the meaning of y-axis label in Figure 9, and expand explanation of fig.9 in the text.
Reply: Ab 480 nm is a spectrometer wavelength – the information is in MM part
Reviewer 4 Report
This paper is a study of the "Preparation and properties of bulk and porous Ti-Ta-Ag biomedical alloys"
The paper is poor, the English must be improved.
- In the „Abstract”, please rephrase: Porosity, morphology, mechanical properties as well as biocompatibility and antibacterial behavior were investigated and related to preparation procedures.
- In the „Abstract”, please correct: The new materials showed a positive behavior in the MTT assay as well as and antibacterial properties. Such materials could find application in the production of hard tissue implants.
- In the „Introduction”, [1-4], which could results in bone resorption which in the worst scenario may lead to implant loosening or bone fracture à which could result in bone resorption, which may lead to implant loosening or bone fracture in the worst scenario
- In the „Materials and Methods”, please make a table of content of your experiments; it will be easier to read.
- In the „Materials and Methods”, please make subtitles at Microstructural and morphological analysis, Corrosion resistance analysis, Wetting and surfaces free energy analysis, Nanoindentation Test (suggestion: Tribolocical results), MTT assay, Antibacterial properties,
- In the „Results and Discussion”, please indicate in the Figures the best result of your experiments and comment.
- Please improve the Reference with current papers.
I recommend major revision.
Author Response
We would like to express our great appreciation to the Editor and respected Reviewers for examining the manuscript carefully and providing us valuable comments and constructive suggestions which are very important for the revision of the manuscript and improving the quality of the work. We have tried our best to revise our manuscript according to their suggestions.
Replies to Reviewer #4:
Reviewer #4:
Comment 1: In the „Abstract”, please rephrase: Porosity, morphology, mechanical properties as well as biocompatibility and antibacterial behavior were investigated and related to preparation procedures.
Reply: improved
Comment 2: In the „Abstract”, please correct: The new materials showed a positive behavior in the MTT assay as well as and antibacterial properties. Such materials could find application in the production of hard tissue implants.
Reply: improved
Comment 3: In the „Introduction”, [1-4], which could results in bone resorption which in the worst scenario may lead to implant loosening or bone fracture à which could result in bone resorption, which may lead to implant loosening or bone fracture in the worst scenario
Reply: improved
Comment 4: In the „Materials and Methods”, please make a table of content of your experiments; it will be easier to read.
Reply: in our opinion another table will be unnecessary copying information from the text. After consultation we decided to not include extra table.
Comment 5: In the „Materials and Methods”, please make subtitles at Microstructural and morphological analysis, Corrosion resistance analysis, Wetting and surfaces free energy analysis, Nanoindentation Test (suggestion: Tribolocical results), MTT assay, Antibacterial properties,
Reply: improved
Comment 6: In the „Results and Discussion”, please indicate in the Figures the best result of your experiments and comment.
Reply: improved
Comment 7: Please improve the Reference with current papers.
Reply: improved
Round 2
Reviewer 1 Report
The authors have made a great work on improving the manuscript text. I will recommend this paper for publication.
Author Response
Once again, we would like to express our great appreciation to the Editor and respected Reviewers for examining the manuscript carefully and providing us valuable comments and constructive suggestions.
Reviewer 2 Report
The authors took into account the Reviewer's comments and improved the quality of the paper
Author Response

(The authors gave the same response as above.)

Reviewer 3 Report
I recommend a fine or minor spell check, e.g., line 175 "propoerties", l.225 "consistingof", l.249 "Table1. along"; l.310 "respectively.. The micrographs".
Author Response
Once again, we would like to express our great appreciation to the Editor and respected Reviewers for examining the manuscript carefully and providing us valuable comments and constructive suggestions.
Replies to Reviewer:
Reviewer Comment 1: I recommend a fine or minor spell check, e.g., line 175 "propoerties", l.225 "consistingof", l.249 "Table1. along"; l.310 "respectively.. The micrographs".
Reply: improved – we check the spelling
Reviewer 4 Report
The revised manuscript fulfills all the reviewer requirements.
I recommend to be accepted as it is.
Author Response

(The authors gave the same response as above.)
